# A Novel Detection Procedure for Mutations in the 23S rRNA Gene of Macrolide-Resistant *Mycoplasma pneumoniae* with Two Non-Overlapping Probes Amplification Assay

**DOI:** 10.3390/microorganisms12010062

**Published:** 2023-12-28

**Authors:** Liyong Liu, Caixin Xiang, Yiwei Zhang, Lihua He, Fanliang Meng, Jie Gong, Jie Liu, Fei Zhao

**Affiliations:** 1National Key Laboratory of Intelligent Tracking and Forecasting for Infectious Diseases, National Institute for Communicable Disease Control and Prevention, Chinese Center for Disease Control and Prevention, Beijing 102206, China; liuliyong@icdc.cn (L.L.); xiangcaixin129@163.com (C.X.); 17639124810@163.com (Y.Z.); helihua@icdc.cn (L.H.); mengfanliang@icdc.cn (F.M.); gongjie@icdc.cn (J.G.); liujie___666@163.com (J.L.); 2School of Public Health, China Medical University, Shenyang 110122, China

**Keywords:** *Mycoplasma pneumoniae*, macrolide resistance, 23S rRNA, real-time PCR

## Abstract

*Mycoplasma pneumoniae* is a significant cause of community-acquired pneumonia, which is often empirically treated with macrolides (MLs), but, presently, resistance to MLs has been a matter of close clinical concern. This assay is intended to contribute to resistance detection of *M. pneumoniae* in clinical practice. A novel real-time PCR assay with two non-overlapping probes on the same nucleic acid strand was designed in this study. It could effectively detect all mutation types of *M. pneumoniae* in 23S rRNA at loci 2063 and 2064. The results were determined by the following methods: ΔCT < 0.5 for MLs-sensitive *M. pneumoniae*; ΔCT > 2.0 for MLs-resistant *M. pneumoniae*; 10 copies as a limit of detection for all types. For detection of *M. pneumoniae* in 92 clinical specimens, the consistency between the results of this assay and the frequently used real-time PCR results was 95.65%. The consistency of MLs resistance results between PCR sequencing and this assay was 100% in all 43 specimens. The assay could not only cover a comprehensive range of targets and have high detection sensitivity but is also directly used for detection and MLs analysis of *M. pneumoniae* in specimens.

## 1. Introduction

*Mycoplasma pneumoniae* (*M*. *pneumoniae*) is an important cause of respiratory tract infections (RTI) in humans, accounting for 10% to 30% of community-acquired pneumonia (CAP) annually, particularly in children [1,2]. A decade-long multi-central study showed that *M*. *pneumoniae* accounted for 19% of acute bacterial respiratory infections in China, particularly up to 59% of acute bacterial respiratory infections in school-age children [3]. For the clinical treatment of *M*. *pneumoniae*, macrolides (MLs) are used as the first-line therapy, especially in children [4]. The in vitro macrolide antibiotic resistance rate of *M*. *pneumoniae* varies greatly worldwide. MLs’ resistance rates of *M*. *pneumoniae* are lower in Europe and the United States but higher in Asia [5,6,7,8,9]. Many studies have shown that patients infected with MLs-resistant *M*. *pneumoniae* (MRMP) strains have a longer duration of clinical symptoms than those infected with MLs-sensitive *M*. *pneumoniae* (MSMP) strains, with children more likely to develop refractory *M*. *pneumoniae* pneumonia (RMPP) [10,11].

The resistance mechanism of *M*. *pneumoniae* is that point mutations in the 23S rRNA at 2063, 2064, or 2617 loci are associated with resistance to macrolides [12]. It was shown that mutations at 2063 and 2064 loci were associated with high levels of MLs resistance, whereas mutation at 2617 locus was associated with low levels of MLs resistance [13]. Studies have shown that the most common mutation is A2063G, which accounts for 93.7% of all mutations, and the three mutations regarding A2063G, A2063T, and A2064G could cover more than 99% of the mutations, but A2063C, A2064C, and C2617G have only been reported individually [14]. Due to the complexity and time-consuming characteristics of *M*. *pneumoniae* antibiotic sensitivity tests, macrolide-resistant detection relies mainly on nucleic acid technology for the detection of mutation sites. At present, the common detection techniques are as follows: sequencing [15], PCR with restriction fragment length polymorphism (RFLP) analysis [16], Cycleave PCR [17], real-time PCR with high-resolution melt analysis (HRMA) [18], MGB/LNA real-time PCR [19], and recently newly reported PNA-LAMP and quenching probe PCR techniques [20]. However, these techniques all have shortcomings. The sequencing technique is labor-intensive. RFLP analysis and allele-specific PCR have the disadvantage of insufficient rapidity, and HRMA requires high performance requirements for real-time PCR instruments. Cycleave PCR, PNA-LAMP, and MGB/LNA real-time PCR can only detect partial mutation types of A2063G and A2064G, and quenching probe PCR requires specific instruments.

To our knowledge, the resistance rate of *M*. *pneumoniae* to macrolides has been high in China [21,22,23]. Currently, the reported macrolide resistance types can cover a wide range of mutation types at 2063 and 2064 loci, but no 2617-mutant-resistant strains have been found [21,22,23,24,25]. In this study, we developed a novel quantitative mutation detection technique with double probes that, on the same nucleic acid strand, covers the full mutation types of *M*. *pneumoniae* at 2063 and 2064 loci, which can be used for identification and macrolide-resistant detection of *M*. *pneumoniae* in clinical specimens.

## 2. Materials and Methods

### 2.1. Ethics Statement

The study was approved by the Ethics Committee of the National Institute for Communicable Disease Control and Prevention, Chinese Center for Disease Control and Prevention (Beijing, China) (ICDC-2023003).

### 2.2. Strains and Nucleotides

The following American Type Culture Collection (ATCC) reference and clinically isolated strains were used to assess the sensitivity, specificity, lowest detectable limit (LDL): *M. pneumoniae* (ATCC 29342, ATCC 15531, ATCC 29085, ATCC 29343, ICDC P005, ICDC BCH388, ICDC WH034), *M. hominis* (ATCC23114), *M. penetrans* (ATCC55252), *U. urealyticum* (ATCC27618), *M. genitalium* (ATCC33530), and *S. aureus*, *E. coli*, *S. epidermidis*, *S. pneumoniae*, *N. meningitides*, *L. pneumophila*, *H. influenzae*, *M. tuberculosis*, *P. aeruginosa*, *K. pneumoniae* belong to clinical isolates and preserved at ICDC. DNA was extracted using a QIAamp DNA Mini Kit (QIAGEN, Hilden, Germany. 51306).

### 2.3. Dual Fluorescent Probe Mutation Detection System

The primers and probes for point mutations of *M. pneumoniae* 23S rRNA at positions 2063 and 2064 are designed by Primer Express 3.0.1 software (Thermo Fisher, Waltham, MA, USA). Primer F: AATCCAGGTACGGGTGAAGACA, primer R: TGCTCCTACCTATTCTCTACATGATAATG, probe A (covers the 2063 and 2064 mutation loci): VIC-ACGGGACGGAAAGA-MGB, probe B (behind probe A and located in the same template strand): FAM-ACTGTAGCTTAATATTGATCAG-MGB (Sangon Biotech, Beijing, China) (Figure 1). The real-time PCR mixture was prepared in a total volume of 25 μL. Each PCR mixture contains the following per reaction: Solarbio 2× mix 12.5 μL (Solarbio, Beijing, China), primer F (25 μM) 0.5 μL, primer R (25 μM) 0.5 μL, probe A (25 μM) 0.3 μL, probe B (25 μM) 0.1 μL, nucleic-acid free water 10.1 μL, template 1.0 μL. Real-time PCR was performed by Q6 fluorescent PCR instrument (Thermo Fisher, Waltham, MA, USA) under the following conditions: 95 °C for 10 min, followed by 45 cycles of 95 °C for 15 s and 65 °C for 15 s.

### 2.4. Determination of Test Results

The fluorescence threshold limit was manually fine-tuned so that the positive standard ∆CT = |CT_VIC_ − CT_FAM_| was within 0.2. The results were determined by the following methods: ΔCT < 0.5 for MSMP; ΔCT > 2.0 for MRMP; no fluorescent signals were detected for non-*M. pneumoniae*.

### 2.5. Evaluation of Method Sensitivity and Specificity

The DNA nucleic acids from 40 *M. pneumoniae* isolates (including 27 MRMP and 13 MSMP isolates) were isolated and identified in our laboratory for sensitivity testing. The nucleic acids from the 14 pathogen strains described above as well as nucleic acids from 40 clinical *M. pneumoniae*-negative respiratory specimens were used in order to conduct specificity testing.

### 2.6. Real-Time PCR Standard Curve and LDL

For all 23S rRNA mutation types of *M. pneumoniae*, the concentration gradient nucleic acids from four strains (ATCC 29342 as the sensitive template, ICDC P005 as the A2063G template, ICDC BCH388 as the A2064G template, ICDC WH034 as the A2063T template) and two synthesized plasmids (the A2063C and A2064C mutations, which have no isolates in China, were constructed by using the pUC57 plasmid) were used to detect the standard curve and LDL. Each nucleic acid was quantified using Qubit 4.0 (Thermo Fisher, Waltham, MA, USA). For a standardized dilution series of quantified nucleic acids, each mentioned above was detected by this assay from 10^7^ copies to 1 copy.

### 2.7. Analysis of MRMP and MSMP Mixed Infection Detection

Quantified ICDCP005 (A2063G mutation) and ATCC29342 (no mutation) nucleic acid were mixed in the proportions of 1%, 5%, 10%, 20%, 50%, 80%, 90%, and 99% to prepare different proportions to test this assay for its ability to identify the mixed MRMP and MSMP infections. Quantified ATCC 15531 (no mutation) and ATCC29342 (no mutation) nucleic acid were mixed in the proportions of 10%, 50%, and 99% to prepare different proportions to test the ability to identify the mixed different MSMP infections. Quantified ICDCP005 (A2063G mutation) and BCH388 (A2064G mutation) nucleic acid were mixed in the proportions of 10%, 50%, and 99% to prepare different proportions to test the ability to identify the mixed different MRMP infections. Two rare clinical specimens [26] with mixed MRMP and MSMP infections (SP01 specimens about 30% MSMP; SP13 specimens about 5% MSMP) were selected to verify this assay.

### 2.8. Clinical Specimen Validation of the Assay

A total of 92 nucleic acids from clinical specimens were selected, which were kept in our laboratory and had undergone *M. pneumoniae* nucleic acid testing, isolation and culture, and sequencing of 23S rRNA gene. The assay was used for *M. pneumoniae* identification and MLs resistance detection as well as comparing the consistency of the results with those already available. The nucleic acids of specimens were detected by real-time PCR method reported in the literature [27]. After culture-positive specimens were purified and underwent enrichment culture, the nucleic acids were extracted for 23S rRNA domain V region amplification and sequence determination to analyze macrolide resistance mutations [13].

## 3. Results

### 3.1. Real-Time PCR Standard Curves and LDL

For sensitive strain ATCC29342 and the five types of mutant strains or plasmids, the detection limits of the assay were all 10 copies, and the linear ranges covered at least seven concentration gradients from 10^7^ copies to 10 copies, with R^2^ > 0.99 of the standard curves (Figure 2). Determination based on ΔCT value is as follows: the ΔCT value of ATCC 29342 strains within the linear range is less than 0.5, which can be accurately determined as MSMP; the ΔCT value of all five types of mutants within the linear range is more than 2.0, which can be accurately determined as MRMP (Table 1).

### 3.2. Sensitivity and Specificity of the Assay

All 40 *M. pneumoniae* isolates were positive by the assay, of which 27 isolates with ΔCT values more than 2.0 were identified as MRMP and 13 isolates with ΔCT values less than 0.5 were identified as MSMP. Therefore, the detection sensitivity is 100%. All the detection results for nucleic acids of 14 specificity-verified pathogens and 40 non-*M. pneumoniae* clinical specimens were negative by the assay, with specificity of 100%.

### 3.3. Analysis of MRMP and MSMP Mixed Infection Detection

The results indicate that different types of MSMP and MRMP strains mixed in different proportions did not affect the results (Table 2). When different proportions of MRMP strains were mixed with MSMP strains, the ∆CT values gradually became smaller as the proportion of MSMP strains increased. If the proportion of MSMP strains is less than 20% and the ΔCT value is more than 2.0, the result is incorrectly judged as a single MRMP in general. If the proportion of MSMP strains is more than 90% and the ΔCT value is less than 0.5, the result is incorrectly judged as a single MSMP strain in general. Moreover, strains with a percentage of MSMP strains in the range of 20% to 90% and a test result of 0.5 < ΔCT < 2.0 are judged to be a mixed sensitive and resistant infection. Clinical specimen SP01 was correctly judged as mixed infection of MSMP and MRMP type for ΔCT = 1.83, but clinical specimen SP13 was incorrectly classified as MRMP for ΔCT = 3.35.

### 3.4. Clinical Specimen Validation of the Assay

The positivity rate of *M. pneumoniae* was 77.17% (71/92) by the assay and 79.25% (73/92) by reported real-time PCR assay in 92 specimens. The consistency of detection between the two methods was 95.65% (88/92). Among them, 47 specimens were positive for *M. pneumoniae* culture, and the mutant results of 23S rRNA amplification and sequence were in 100% (47/47) agreement with the results of this method (Appendix A).

## 4. Discussion

Macrolide antibiotics are the drugs of first choice for *M. pneumoniae* infections, and the issue of their resistance has been a clinical concern. The mechanism of resistance has been defined as in vitro resistance to macrolide antibiotics triggered by mutations in the 23S rRNA at 2063, 2064, or 2617 loci, in which mutations 2063 and 2064 cover more than 99.5% of all the mutation types [14]. Currently, the most common *M. pneumoniae* macrolide resistance detection technique is PCR, relying on a variety of probes, which is highly sensitive but only targets two main mutation types, A2063G and A2064G [20,28,29].

In this assay, the unique feature is to design two fluorescent probes on the same nucleic acid strand, one of which covers the mutated region and serves as a mutation detection probe (probe A), and the other is located a few bases later and serves as an indicator probe (probe B) that serves as an identification function of *M. pneumoniae* as well as an indicator reference about presence of mutation. The two probes are located on the same nucleic acid chain with no overlap in positions. When an MSMP strain was detected, both probes would bind specifically to their respective targets, and it was almost identical regarding the fluorescence signals released by hydrolysis of the two probes, which were presented by the high degree of concordance in the fluorescence curves of the FAM and VIC for different nucleic acid concentrations in the exponential phase of amplification, showing a very small difference in the CT value. However, for detection of mutant isolates, the binding ability of probe A was drastically reduced, while the binding ability of probe B was unchanged due to the absence of mutation at the target site, which was shown by the large CT difference between the FAM and VIC fluorescence curves for different nucleic acid concentrations of various mutant isolates. Probe B is an internal reference probe that was cleverly designed with probe A in the same PCR amplification assay, which has more internal reference value than making them into different amplification assays and avoiding internal reference fluctuation caused by the efficiency difference between different amplification assays. Moreover, because the positions of the two probes do not overlap, there is no effect of mutual antagonism or competition causing a decrease in detection sensitivity.

Theoretically, when a mutation is present, the mutation detection probe (probe A) will not bind the template due to a single base mismatch with the template strand. However, the mutation detection probe modified with MGB still produces non-exponential amplification due to non-specific binding [30], which is reflected in the amplification curve as pushback of the ∆CT value or the lack of an exponential phase of the amplification curve showing only a slow upward trend, which results in a significant decrease in the fluorescence signal value at the end of the amplification (Figure 2). This suggests that, in the absence of antagonistic probes, a single MGB probe cannot completely avoid non-specific amplification. In addition, at present, the probe positions for wild-type and mutant strain in common point mutation detection technologies also almost overlap [17,19,20,29], which has the advantage that the results are easy to determine through the competition inhibition and can almost eliminate non-specific amplification caused by the other probe. However, to a certain extent, the competition between the two probes’ antagonism could reduce the sensitivity of the detection, which leads to the disadvantage that the CT values shift backward and the detection types are covered incompletely. In this assay, an indicator probe at different positions on the same template chain is designed, which solves the disadvantage of antagonistic probes, but the results need to be determined according to ∆CT.

The results of the standard curve show that this assay could not only detect all mutation types of 23S rRNA 2063 and 2064 loci but also has a good linearity in the range from 10^7^ to 10 copies with probe B, which covers the range of *M. pneumoniae* carrying capacity in various clinical specimens. Meanwhile, this assay could accurately detect and quantitatively analyze specimens with different carrying capacity. The detection limit of this assay is about 10 copies, which is close to the detection limit of the currently used fluorescence PCR method for *M. pneumoniae*. Through 92 clinical specimens [21,24,25] comparative testing, it was found that the consistency of the method was as high as 95.65% (88/92) with the existing *M. pneumoniae* real-time PCR assay [27]. At the same time, not only did this assay have highly sensitive detection ability, but also for the consistency of macrolide resistance analysis and the consistency of the conventional PCR sequencing method [13], the results are all 100% (47/47), which solves the disadvantage that many clinical specimens encounter difficulty in performing 23S rRNA gene detection directly by traditional amplification techniques due to low load of *M. pneumoniae*.

As an acute respiratory pathogen, clinical cases of *M. pneumoniae* are usually considered to be single-isolate infection, while mixed infections are very rare [26]. Currently, all the mutation detection methods for *M. pneumoniae* are based on this feature [16,17,18,19,20,28,29], so they have limited discriminatory power to detect a mixture of MSMP and MRMP. In this study, different MRMP strains or MSMP strains mixed in different proportions did not affect the results (Table 2). However, when different proportions of MSMP strains (ATCC 29342) were mixed with MRMP (ICDCP005) strains, the results showed that the ∆CT value becomes smaller with the increase in the proportion of MSMP strains. When the proportion of MSMP strains is in the range of 20% to 90% and the test result is 0.5 < ΔCT < 2.0, this could be judged as a mixed infection of sensitive and resistant *M. pneumoniae*. However, when beyond the range of mixed infections, this method cannot distinguish between what kind of infection. A good example is that mixed infection clinical specimen SP01, with the proportion of MSMP strains close to 30% and the detection ΔCT = 1.83, was accurately determined to be a mixed infection of sensitive and resistant. However, for mixed infection clinical specimen SP13, the proportion of MSMP strains was very low, with only about 5%, and the detection ΔCT = 3.35, which was incorrectly determined to be a macrolide-resistant type. These show that this assay is similar to all the existing nucleotide mutation detection methods of *M. pneumoniae*, which is not suitable for the detection of mixed infection specimens. Fortunately, mixed different isolate infections regarding *M. pneumoniae* are rarely reported.

In conclusion, we established a novel technique for the macrolide resistance detection of *M. pneumoniae* by taking advantage of the two completely non-overlapping adjacent indicator probes and mutation probes designed on the same target sequence strand. Not only could it be suitable for the direct detection of *M. pneumoniae* in nucleic acids from clinical specimens and cover all mutation types at 2063 and 2064 loci but it also has the characteristics of high sensitivity and specificity as well as quantitative analysis for detection results. Therefore, this is a technique worth promoting and applying to *M. pneumoniae* detection and macrolide resistance analysis with clinical specimens.

## Figures and Tables

**Figure 1 microorganisms-12-00062-f001:**
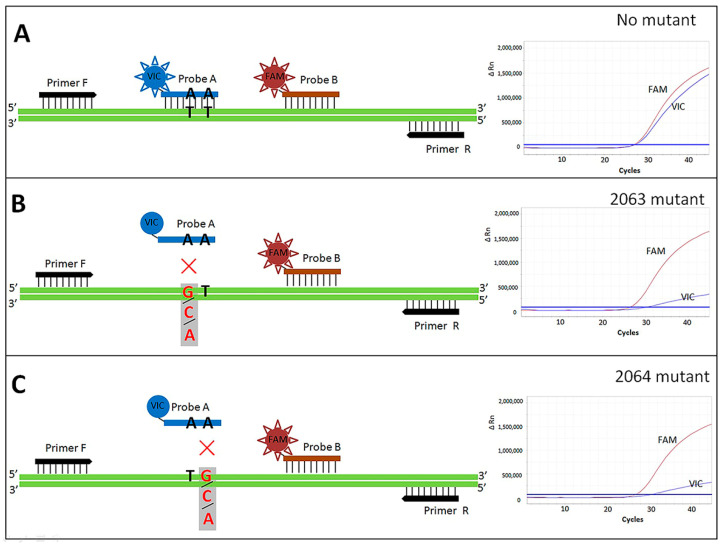
Schematic diagram for detection principle of the assay. (**A**) Detection of templates regarding wild-type (MSMP) strains; (**B**) detection of mutant strains at locus 2063; (**C**) detection of mutant strains at locus 2064. Note: the fluorescent signal is generated by enzymatic hydrolysis after the probe matches the target sequence, which is only for demonstration in the figure.

**Figure 2 microorganisms-12-00062-f002:**
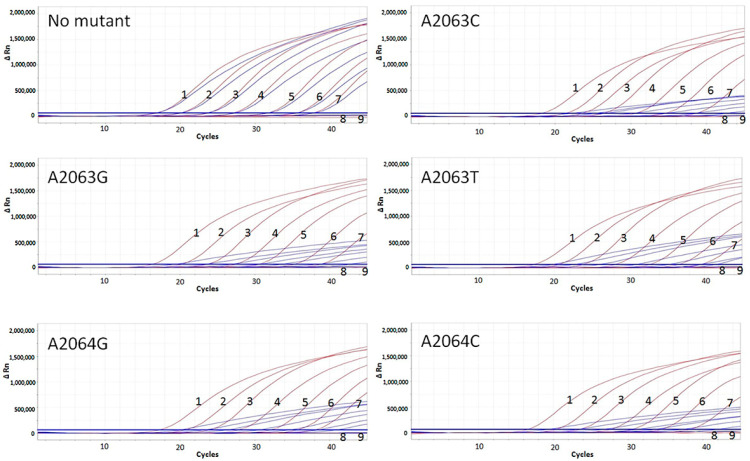
Standard curve of detection regarding concentration gradient for MSMP (ATCC 29342) and different mutation types of MRMP. The red color represents FAM fluorescence for *M. pneumoniae* determination. The blue color represents VIC fluorescence for mutation site determination of *M. pneumoniae*. 1: 10^7^ copies; 2: 10^6^ copies; 3: 10^5^ copies; 4: 10^4^ copies; 5: 10^3^ copies; 6: 10^2^ copies 7: 10 copies; 8: 1 copy; 9: NTC.

**Table 1 microorganisms-12-00062-t001:** Detected CT and ∆CT values for various mutation types of *M. pneumoniae* 23S rDNA under different copy number template amounts.

Nucleic Acid Gradient (Copies)	CT and ΔCT Values																	
	ATCC29342 (A2063)			Plasmid (A2063C)			ICDC BCH388 (A2064G)			ICDC WH034 (A2063T)			ICDC P005 (A2063G)			Plasmid (A2064C)		
	FAM	VIC	ΔCT	FAM	VIC	ΔCT	FAM	VIC	ΔCT	FAM	VIC	ΔCT	FAM	VIC	ΔCT	FAM	VIC	ΔCT
10^7^	16.96	16.69	0.28	18.28	22.11	3.83	17.22	19.99	2.77	17.26	19.62	2.36	16.26	19.18	2.92	16.55	19.25	2.70
10^6^	20.12	20.06	0.05	21.42	25.25	3.83	20.80	23.22	2.42	20.36	22.50	2.14	20.22	23.28	3.07	20.33	23.22	2.89
10^5^	23.76	23.55	0.22	23.56	26.59	3.03	23.98	26.27	2.29	23.85	25.98	2.13	23.74	26.83	3.09	23.58	26.58	3.00
10^4^	27.09	27.31	0.22	26.45	30.30	3.85	27.86	30.48	2.62	28.09	30.76	2.67	27.43	30.36	2.93	27.61	31.23	3.62
10^3^	31.71	31.79	0.09	30.65	34.59	3.94	32.17	34.59	2.42	32.63	35.15	2.51	30.60	33.61	3.01	30.49	33.40	2.91
10^2^	35.60	35.46	0.14	34.60	38.30	3.70	35.70	38.34	2.65	36.49	39.40	2.91	35.32	38.35	3.04	34.37	37.13	2.75
10	37.72	37.93	0.20	38.63	42.59	3.96	38.03	40.85	2.82	39.54	41.64	2.10	38.83	42.58	3.75	38.19	41.85	3.66
1	Neg	Neg	Neg	Neg	Neg	Neg	Neg	Neg	Neg	Neg	Neg	Neg	Neg	Neg	Neg	Neg	Neg	Neg
NTC	Neg	Neg	Neg	Neg	Neg	Neg	Neg	Neg	Neg	Neg	Neg	Neg	Neg	Neg	Neg	Neg	Neg	Neg

**Table 2 microorganisms-12-00062-t002:** Results of detection for different types of macrolide antibiotic mutations mixed with sensitive *M. pneumoniae* strains.

Proportion	ATCC 29342: ICDC P005 (No Mutant): (A2063G)			ATCC 29342: ATCC15531 (No Mutant): (No Mutant)			ICDC BCH388: ICDC P005 (A2064G): (A2063G)		
	FAM	VIC	ΔCT	FAM	VIC	ΔCT	FAM	VIC	ΔCT
99:1	28.38	28.63	0.25	NT	NT	NT	NT	NT	NT
90:10	28.42	28.82	0.40	27.21	27.48	0.27	27.21	29.95	2.74
80:20	28.07	29.03	0.96	NT	NT	NT	NT	NT	NT
50:50	27.88	29.53	1.65	27.33	27.41	0.08	27.77	30.59	2.82
20:80	27.98	30.13	2.15	NT	NT	NT	NT	NT	NT
10:90	27.46	30.84	3.38	27.69	28.01	0.32	27.85	30.72	2.87
1:99	27.42	31.41	3.99	NT	NT	NT	NT	NT	NT

NT: not detected.

## Data Availability

Data are contained within the article.

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
