# Peer review of "A Novel Detection Procedure for Mutations in the 23S rRNA Gene of Macrolide-Resistant *Mycoplasma pneumoniae* with Two Non-Overlapping Probes Amplification Assay"

_microorganisms, 2023, doi:10.3390/microorganisms12010062_

Round 1

Reviewer 1 Report

Comments and Suggestions for Authors

The article’ content not only is informative with clearly articulated ideas and comprehensive data, but also introduce new methods as well as new ideas in the research techniques of respiratory infectious disease, which is a boost for Mycoplasma pneumoniae research. At the same time, the establishment of Mycoplasma pneumoniae dual-probe multi-locus drug resistance gene detection technology in clinical specimens, which solves the shortcomings that currently poor sensitivity of gene detection and incomplete coverage of the pain points in the drug resistance of Mycoplasma pneumoniae and plays an important role in the field of infectious diseases.

 However, the following minor points need to be considered.

1Introduction, in accordance with the International Code of Nomenclature of Bacteria: Bacteriological Code, if a series of species names all belonging to the same genus, it is customary to abbreviate the name of the genus in all but the first species, even if it is the first mention of the succeeding species (https://www.ncbi.nlm.nih.gov/books/NBK8816/) Therefore, "Mycoplasma pneumoniae (M. pneumoniae)" is incorrect, "Mycoplasma pneumoniae can be written as "Mycoplasma pneumoniae (Mp)” or simply do not abbreviate it. When it appears again, it can be written as “M. pneumoniae”

 2 2.1. Ethics statementdelete one of the parentheses

32.2. Strains and nucleotides the genus name should be abbreviated

42.3. Dual fluorescent probe mutation detection systemuM should be “μM

Author Response

Thanks for the reviewer's suggestions. We have revised the manuscript according to the suggestions.

Reviewer 2 Report

Comments and Suggestions for Authors

The authors describe a method to detect macrolide resistance mutations in Mycoplasma pneumoniae. They have extensively validated the assay and clearly described the methodology. The following minor remarks should be addressed:

-          Authors should elaborate more on how their methodology overcomes the shortcomings of the other techniques mentioned in the introduction. Especially the statement on specific instrument required for HRMA is incorrect.

-          The discrepancies in the clinical specimen validation of the assays should be explored in more detail. For example what are Ct values of the 3 samples positive by real-time PCR but not via the described assay. How to explain the 1 sample that was negative by real-time PCR but was positive by the new method. Also Ct values with regard to culture results are interesting to present.

Author Response

  1. Thanks for the reviewer's suggestion. The description about HRMA was not appropriate in the manuscript. HRMA does not require specific instruments and applications. Its disadvantage is that HRMA requires high performance requirements for real-time PCR instruments. We have revised it in the manuscript.
  2. We think that your question is very good. In fact, different targets are used in different nucleic acid detection methods, which could cause some differences for sensitivity and specificity of detection methods. Moreover, in the detection of low concentration specimens with nucleic acid load close to the detection limit of the method, the detection results of different nucleic acid detection methods may be different, and even the results also could be different in different replicated experiments by the same method. In this study, the consistency of the two nucleic acid methods was 95.65% (88/92). The several specimens you mentioned with inconsistent detection are all specimens with CT value>35, which could lead to inconsistent detection results between the two methods. In general, the CT value of culture-positive specimens is relatively low, but it is not absolute. In this study, for comparison of two nucleic acid detection method we used the POS and NEG but not the CT value. This is because we hope to simplify the content of the results in the table and highlight the determination of △CT results.

Reviewer 3 Report

Comments and Suggestions for Authors

The methods are poorly described.

What is the reproducibility of the results?

In References 12, 19 and 20, correct 23s Rna to 23S rRNA.

In References 16, 17, 18, 27, 29 and 30, correct Pcr to PCR.

Author Response

Thanks for your suggestion. We have revised the manuscript according to the suggestions.

We have done repeated experiment by the methods, which showed that the results had a very good repeatability. But due to space limitation we did not show the results in the manuscript. Two figures are showed as following: the one is that the four repeats of resistant nucleic acid detection and another one is that the two repeats of sensitive nucleic acid amplification curves for different concentration gradients.

Thanks very much. We have revised the spell about 23S rRNA and PCR in the manuscript.